# Eyes on the Image: Gaze Supervised Multimodal Learning for Chest X-ray Diagnosis and Report Generation

## Abstract

Medical vision-language models still struggle to match radiologists' attention and to verbalize findings with explicit spatial grounding. We address this gap with a two-stage multimodal framework for chest X-ray interpretation built on the MIMIC-Eye dataset. In the first stage introduces a gaze-token classifier that fuses image patches, bounding-box masks, transcription embeddings, and radiologist fixations. A curriculum-scheduled, trust-calibrated composite loss supervises the gaze token, boosting both accuracy and spatial alignment. Adding fixation supervision raises AUC 4.4% and F1 13.3%, and Pearson correlation rises to 0.306, confirming clinically relevant focus. In stage 2, classifier predictions are translated into region-specific diagnostic sentences. Confidence-weighted keywords are extracted, mapped to 17 thoracic regions through an expert dictionary, and expanded with a prompted large language model, boosting clinical-term BERTScore and ROUGE scores over keyword baselines. All components are toggle-able for ablation, and the full pipeline is reproducible, offering a new benchmark for interpretable, gaze-aware chest-X-ray analysis. Integrating eye-tracking signals demonstrably enhances both diagnostic accuracy and the transparency of generated reports.

## 1 Introduction

Radiology reports shape clinical decision making: treatment plans, follow-up imaging, and even surgical interventions often depend on the language a radiologist chooses to record Casey et al. (2021); Liu et al. (2019). Accordingly, report-generation systems must be precise as well as capturing subtle pathologies and be explainable, so that every statement can be traced back to verifiable image evidence Tanida et al. (2023). Producing such reports automatically from chest X-rays is therefore both a high-impact goal and a stringent test of multi-modal reasoning Yang et al. (2023).

Yet the underlying data are stubbornly heterogeneous. Pixel-level visual cues, sentence-level textual descriptions, and time-stamped attentional traces collected via eye-tracking each operate on different scales and carry different noise profiles Karargyris et al. (2021); Lanfredi et al. (2022). Aligning these modalities is complicated by (i) reader-specific gaze patterns, (ii) limited bounding-box coverage, and (iii) the need to express findings in radiologist-approved terminology. A successful solution must fuse all three signals without diluting any one of them Ma et al. (2024).

To this end, we present four contributions. An overview of the pipeline is shown in Figure 1.

- *Gaze-Token Guided Multimodal Fusion for Disease Prediction*. A learnable gaze token is injected into the ViT patch stream and modulated by a bounded gating layer; its attention is supervised by a trust-calibrated composite loss combining pixel fidelity (Mean Squared Error (MSE), Kullback-Leibler (KL)), pattern similarity (Pearson correlation), and geometric alignment (normalised Center-of-Mass (COM)). All four terms are weighted by fixation density, entropy, and anatomy masks, then scheduled via a curriculum that increases gaze influence. This unifies spatial precision and alignment while preventing gaze dominance.

- *Quantitative Gaze-Attention Validation*. Controlled ablations show consistent focus on clinically relevant regions: Jensen-Shannon Divergence (JSD) $< 0.45$, Pearson correla-

tion ≈ 0.30, and attention entropy approaches the human upper bound, without sacrificing classification scores.

- *Region-Grounded, Keyword-Driven Report Generation.* Stage 2 converts classifier logits into coherent reports by (i) extracting confidence-weighted diagnostic keywords, (ii) mapping them to 17 canonical thoracic regions, and (iii) prompting an LLM to emit region-conditioned sentences. The pipeline preserves spatial grounding from image to text and improves clinical keyword recall.

- *Modular, Inspectable MIMIC-Eye Pipeline.* Encoders, projection blocks, fusion gates, composite losses, and the report generator are toggle-able via config flags. Intermediate outputs are saved, enabling transparent error analysis, reproducible ablations, and easy integration with other public medical-imaging datasets.

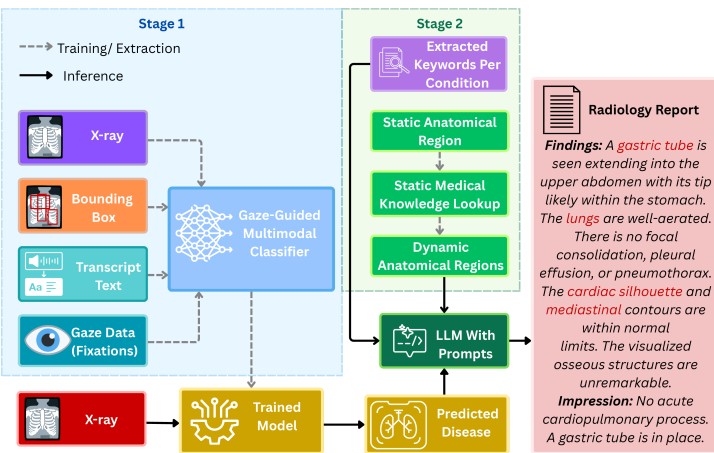

Figure 1: Overview of the proposed multi-modal pipeline. Radiograph, bounding-box masks, transcripts, and eye-tracking fixations are aligned through a contrastive objective; at inference, it takes only the radiograph modules to generate region-grounded radiology reports.

## 2 RELATED WORK

Recent medical vision–language models couple large-scale chest-X-ray corpora with transformer backbones to align image embeddings and report tokens Lu & Wang (2025); You et al. (2023a). Approaches such as MedCLIP Wang et al. (2022), BioViL Bannur et al. (2023a), and Llama-Med pre-train with paired (image, sentence) contrastive objectives and subsequently fine-tune for tagging or report generation, demonstrating strong zero-shot transfer to unseen pathologies Zhang et al. (2023). Although effective at global alignment, these methods operate on whole-image/whole-sentence pairs and provide limited guidance on where in the image a predicted phrase originates Li et al. (2025). Parallel efforts leverage gaze traces as an auxiliary supervisory signal: fixation maps are injected either as soft attention masks or as auxiliary channels, encouraging the encoder to focus on diagnostically salient regions without requiring extra pixel-level labels Ma et al. (2024); Wang et al. (2024).

Complementary to vision–language alignment, keyword and region- aware generators explicitly ground narrative statements in anatomic sub-regions Tanida et al. (2023); Chen et al. (2024). A complementary direction prompts LLMs with gaze and region cues to steer generation without re-training (Kim et al., 2025). Pipelines such as MS-CXR Boecking et al. (2024) and REFLACX Lanfredi et al. (2022) first predict diagnostic keywords, then slot them into structured prompt conditioned on pre-computed bounding boxes, yielding reports with higher factual correctness. These frameworks, however, depend on accurate region detectors and omit attentional cues. Finally, contrastive and multimodal fusion techniques combine heterogeneous inputs; images, clinical labels, bounding boxes, and gaze sequences within a unified representation space; InfoNCE (Information Noise-Contrastive Estimation) style losses balance the modalities while late-fusion transformers aggregate their features Liu et al. (2021); Hayat et al. (2022). Our work intersects these strands

by integrating gaze-guided contrastive learning with a region-grounded, keyword-driven generator, thereby coupling attentional supervision with spatially explicit report generation.

# 3 METHODOLOGY

## 3.1 OVERVIEW AND MOTIVATION

Our goal is to build a multimodal fusion architecture that integrates four complementary information sources encountered during chest X-ray interpretation: (1) the chest X-ray; (2) binary masks that mark anatomically defined bounding boxes; (3) the radiologist's transcription; and (4) eye-tracking fixation sequences recorded during the radiologist's reading. We fuse these four streams with a gaze-token Vision Transformer. Image patches, mask projections, sentence embeddings, and fixation embeddings are projected to 768-dimension tokens, concatenated, and processed by multi-head self-attention. A learnable gaze token attends to image-patch tokens through a bounded gating layer, letting human gaze steer but not dominate-internal attention.

Unlike prior pipelines, we incorporate eye-tracking data as an auxiliary supervisory signal, offering fine-grained attentional guidance without explicit localization labels. Pre-rendered bounding box masks, introduce broad spatial priors without requiring dense supervision; bridging the gap between image space and semantic concepts. Together, these components yield features that transfer well across datasets and support interpretable downstream generation tasks.

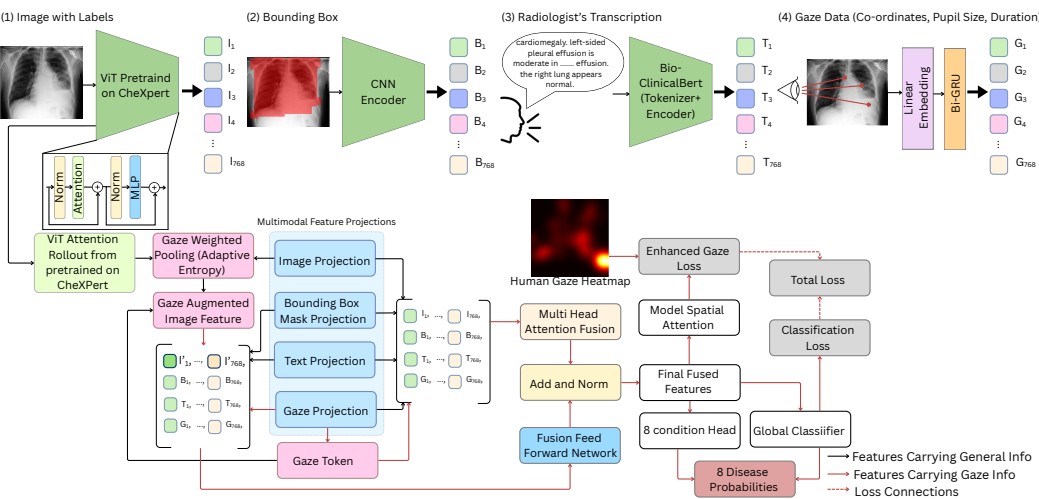

Figure 2: Multimodal chest-X-ray classifier (Stage 1): image, bounding-box, text, and gaze features are independently encoded, concatenated, refined by a cross-modal attention block, and passed through an MLP fusion network whose global and condition-specific heads are ensembled to predict eight disease labels.

## 3.2 INPUT MODALITIES AND CONTRASTIVE LEARNING

Figure 2 gives an overview of the four encoders. Unless noted otherwise, all projected feature vectors have dimensionality, $d = 768$.

**Radiograph (img).** Each $224 \times 224$ radiograph is processed by a ViT-BASE Dosovitskiy et al. (2020) backbone that has been transfer-learned on CheXpert Irvin et al. (2019). From the class token embedding (CLS) token $\mathbf{z}_{img} \in R^{768}$ we derive the image embedding with a two-layer projection head:

$$\mathbf{h}_{img} = \mathrm{LN}\left(W_2 \, \mathrm{GELU}\left(\mathrm{LN}(W_1 \mathbf{z}_{img})\right)\right), \tag{1}$$

where $W_1, W_2$ are learned linear maps, LN denotes Layer Normalization and GELU denotes Gaussian Error Linear Unit. A Dropout (0.15) layer follows each linear transformation in the high-capacity ("enhanced") configuration.

**Bounding-box mask (*bbox*).** The union of all reader-annotated boxes is rasterised into a binary $224 \times 224$ mask and passed through a lightweight convolutional encoder built from stacked CONV+NORM+GELU+ blocks with intermittent $2 \times 2$ pooling. The resulting feature map is global-average-pooled and linearly projected to dimension $d$, producing the embedding $\mathbf{h}_{\text{bbox}}$. This channel supplies a coarse anatomical prior without requiring dense supervision.

**Report text (*text*).** Transcripts are encoded by BioClinicalBERT Alsentzer et al. (2019). We freeze the backbone and project the `[CLS]` representation with $z_{\text{text}}$,

$$\mathbf{h}_{\text{text}} = W_4 \, \text{GELU} \left( \text{LN}(W_3 \mathbf{z}_{\text{text}}) \right), \tag{2}$$

followed by LN and Dropout, matching the implementation.

**Fixation sequence (*gaze*).** A variable-length sequence of $(x, y, \Delta t, \text{pupil})$ tuple is first embedded to 64 dimensions and then encoded by a bi-directional gated recurrent unit (bi-GRU). The enhanced model employs two layers with hidden size 384, giving a 1,536-dimensional concatenated state that is linearly projected to $\mathbf{h}_{\text{gaze}}$.

### 3.2.1 GAZE–MODEL VARIANTS: BASELINE VS. ENHANCED

We instantiate two configurations that differ solely in how they exploit fixation data.

**Baseline gaze.** Fixation tuples are encoded by a bidirectional GRU (bi-GRU) and temporally pooled to a vector $\mathbf{h}_{\text{gaze}}$, which is concatenated with the other modality features. This variant produces no attention map and is trained only with the classification loss $\mathcal{L}_{\text{cls}}$ (and optional image-text InfoNCE $\mathcal{L}_{\text{ITC}}$), so gaze influences the model solely through the fused feature.

**Enhanced gaze.** The enhanced configuration introduces three mechanisms: **(1) Explicit attention maps.** A lightweight decoder converts the fused token into a $14 \times 14$ heat map $\mathbf{A}_{\text{model}}$ (one value per ViT patch). For supervision or visualization this map can be bilinearly up-sampled to $28 \times 28$ and, if needed, to $224 \times 224$; **(2) Trust-calibrated supervision.** Raw fixations are smoothed into $\mathbf{A}_{\text{gaze}}$. A composite loss: MSE, KL divergence, Pearson correlation, and centre-of-mass aligns $\mathbf{A}_{\text{model}}$ with $\mathbf{A}_{\text{gaze}}$; each term is modulated by fixation density, entropy, and anatomy masks and introduced by a curriculum scheduler (Eq. 6); **(3) Cross-modal consistency.** An InfoNCE term (Eq. 5) aligns image and text embeddings, reinforcing semantic agreement between visual regions and the report transcript; no dedicated gaze–text contrastive loss is used.

These additions guide the network to reproduce radiologists' visual search patterns rather than treat fixation statistics as auxiliary features.

### 3.3 FUSION AND PREDICTION

**Token sequence.** The ViT yields a gaze-augmented image embedding $\mathbf{h}_{\text{img}}$. Patch attentions are rolled out and re-pooled with fixation-entropy weights before projection. Bounding-box, text, and gaze branches produce $\mathbf{h}_{\text{bbox}}, \mathbf{h}_{\text{text}}, \mathbf{h}_{\text{gaze}}$. In this context X is the concatenated feature vector that collects the four modality embeddings before any further attention or MLP processing:

$$\mathbf{X} = \left[ \mathbf{h}_{\text{img}} \, \| \, \mathbf{h}_{\text{bbox}} \, \| \, \mathbf{h}_{\text{text}} \, \| \, \mathbf{h}_{\text{gaze}} \right] \in \mathbb{R}^{3072}.$$

When enabled, a learnable 768-d *gaze token* gates information into $\mathbf{h}_{\text{img}}$ in the next step and is not kept as a separate token.

**Attention/gating.** $\mathbf{X}$ is refined by one lightweight attention-gating block (8 heads, GELU, dropout 0.15); otherwise the block is skipped.

**Fusion MLP and classifier.** The (optionally refined) 3072-d vector passes through a two-layer MLP ($3072 \rightarrow 1536 \rightarrow 768$, GELU, dropout 0.15) to form $\mathbf{h}_{\text{fusion}}$, which feeds a global sigmoid head that outputs eight disease logits $\ell \in \mathbb{R}^8$.

This lets fixation cues steer the image embedding via gaze-weighted pooling and the optional gaze token while keeping the classifier simple and parameter-efficient.

## 3.4 TRAINING OBJECTIVES

**Classification loss.** Multi-label focal loss $L_{\text{cls}}$ with class-balanced positives supervises the eight disease labels.

**(2) InfoNCE contrastive loss.** For image–gaze and image–text pairs we minimize the InfoNCE objective van den Oord et al. (2019), derived from noise-contrastive estimation Gutmann & Hyvärinen (2012):

$$\mathcal{L}_{\text{NCE}} = -\frac{1}{|P|} \sum_{(i,j) \in P} \log \frac{\exp(\text{sim}(\mathbf{h}_i, \mathbf{h}_j)/\tau)}{\sum_k \exp(\text{sim}(\mathbf{h}_i, \mathbf{h}_k)/\tau)}, \tag{3}$$

where $\text{sim}(\mathbf{u}, \mathbf{v}) = \frac{\mathbf{u}^\top \mathbf{v}}{\|\mathbf{u}\|\|\mathbf{v}\|}$ and $\tau = 0.07$.

**Proposed Composite gaze-alignment loss.** Let $A_{\text{model}} \in \mathbb{R}^{224 \times 224}$ be the decoded attention map and $A_{\text{gaze}}$ the fixation heat map:

$$L_{\text{gaze}} = w_q \Big( \underbrace{\|A_{\text{model}} - A_{\text{gaze}}\|_2^2}_{\text{MSE}} + \underbrace{\text{KL}\big(\sigma(A_{\text{gaze}}) \,\|\, \sigma(A_{\text{model}})\big)}_{\text{KL}} + \underbrace{1 - \rho\big(A_{\text{model}}, A_{\text{gaze}}\big)}_{\text{Corr}} + \underbrace{\frac{\|\text{CoM}(A_{\text{model}}) - \text{CoM}(A_{\text{gaze}})\|_2}{224\sqrt{2}}}_{\text{CoM}} \Big),$$
$$\tag{4}$$

where $w_q = \sqrt{N_{\text{fix}}}\, q_{\text{score}}$ weighs samples by fixation density and quality; $\sigma$ is softmax.

**Total loss.**

$$L = L_{\text{cls}} + \lambda_{\text{NCE}} L_{\text{NCE}} + \lambda_{\text{gaze}} L_{\text{gaze}}, \qquad \lambda_{\text{NCE}} = 0.1, \ \lambda_{\text{gaze}} = 0.3. \tag{6}$$

Thus, 60% of the optimisation signal targets label accuracy, 10% enforces cross-modal consistency, and 30% enforces spatial alignment with radiologist gaze. All experiments: AdamW (LR $6 \times 10^{-6}$), batch 32 (8 low-mem), 40 epochs, cosine; "Fine Tune" folds validation into training for a final pass (test unchanged; hyperparameters fixed). For baselines: $5 \times 10^{-6}$, 32, 35; ViT-only: $5 \times 10^{-5}$, 128, 20. Experiments ran on an Intel Core i9-14900K CPU and a single NVIDIA RTX 4090 GPU (24 GB VRAM).

## 3.5 TWO-STAGE KEYWORD EXTRACTION PIPELINE

We extract keywords per condition in two steps. **Stage 1:** Gemini 2.5 Pro reads the full report and the eight target pathologies, then proposes a ranked list for each condition (temperature=0.1, $top-k = 1$). Requests run in mini-batches of 30 with exponential backoff (base 3 s; timeout 120 s) and progress logging. **Stage 2:** A second Gemini pass filters the candidates by removing lexical variants, boilerplate (e.g., "no evidence of"), duplicates, and cross-condition leakage. It outputs a simple YES/NO per keyword; confidence is suppressed so decisions rely on semantic meaning, reducing bias from overconfident errors. On the development set (7,322 keywords), 49.5% are kept and 50.5% dropped. The final vocabulary is compact and precise (about $390 \pm 230$ unique keywords per condition; e.g., Atelectasis 164, Lung Opacity 782, Support Devices 683) and is used for anatomical-region matching and structured report generation.

## 3.6 ANATOMICAL REGION MAPPING AND REPORT GENERATION

We maintain a dictionary of 17 thoracic regions, each represented by a bounding-box tuple $(x_{\min}, y_{\min}, x_{\max}, y_{\max})$ and a list of lexical aliases. During training, EyeGaze/REFLACX boxes are normalized to $[0, 1]$ using the recorded image dimensions $(W, H)$, validated, and robustly aggregated per region; at inference, Gemini-cleaned keywords are matched (case-insensitive, fuzzy similarity) to the alias lists. Successful matches activate binary region flags, yielding a sparse 17-dimensional anatomical mask shared across image, gaze, and text streams. The normalized region bounds are later scaled to a $512 \times 512$ dimension for mask rendering and visualization.

**Normalization, validation, and robust averaging.** For each annotation table we (i) normalize coordinates, (ii) apply a validation gate requiring $0 \le x_1 < x_2 \le 1, 0 \le y_1 < y_2 \le 1$, known $(W, H)$, a valid region identifier, and a confidence score (for REFLACX); and (iii) aggregate all valid boxes per region by an element-wise median. If the median is degenerate (non-positive width/height or area $< \tau_{\text{box}}$), we fall back to a confidence-weighted average, with $b_i \in [0, 1]^4$ and confidences $c_i$ (default $c_i=1$ if absent), returning a mapping $M[r] \in [0, 1]^4$ for each region $r$. The complete procedure is summarized in Algorithm 1.

---

**Algorithm 1** Anatomical Region Bounds Aggregation

---

**Require:** Patient set $\mathcal{P}$; region list $\mathcal{R} = \{r_1, \ldots, r_{17}\}$
**Ensure:** Normalized per-region bounds $M : \mathcal{R} \to [0,1]^4$
1: Initialize $L[r] \leftarrow \emptyset$ for all $r \in \mathcal{R}$
2: **for all** $p \in \mathcal{P}$ **do**
3:     Load image size $(W, H)$ and boxes $\mathcal{B}$
4:     **for all** $(r, x_1, y_1, x_2, y_2, c) \in \mathcal{B}$ **do**
5:         $(x_1, y_1, x_2, y_2) \leftarrow (x_1/W, y_1/H, x_2/W, y_2/H)$
6:         **if** $r \in \mathcal{R}$ and $0 \le x_1 < x_2 \le 1$ and $0 \le y_1 < y_2 \le 1$ **then**
7:             Append $\big((x_1, y_1, x_2, y_2),\, c \textbf{ or } 1\big)$ to $L[r]$
8:         **end if**
9:     **end for**
10: **end for**
11: **for all** $r \in \mathcal{R}$ **do**
12:     $B \leftarrow \{b : (b, c) \in L[r]\}, \quad C \leftarrow \{c : (b, c) \in L[r]\}$
13:     $\bar{b} \leftarrow \text{median}(B)$                               ▷ element-wise
14:     **if** $(\bar{b}_{x_2} - \bar{b}_{x_1} \le \tau_{\text{box}})$ or $(\bar{b}_{y_2} - \bar{b}_{y_1} \le \tau_{\text{box}})$ **then**
15:         $M[r] \leftarrow \dfrac{\sum_i c_i b_i}{\sum_i c_i + \varepsilon}$         ▷ confidence-weighted avg.
16:     **else**
17:         $M[r] \leftarrow \bar{b}$
18:     **end if**
19: **end for**
20: **return** $M$

---

**Report generation.** The classifier outputs posterior probabilities for eight target pathologies. Conditions with p(c) > 0.60 and their activated regions are passed to a Gemini 2.5 Pro prompt that (temperature = 0.3, top-$k$ = 1) injects regional context, enforces radiology style, and produces distinct *findings* and *impression* . The prompt includes strict instructions to avoid unsupported statements. API calls use up to five retries with exponential backoff; on final failure, a concise local fallback paragraph is emitted. We serialize per-condition probabilities, matched keyword sources, and contributing region indices to support interpretability dashboards. This replaces the earlier (unused) phrase-level provenance scheme and preserves the high-recall region mapping while leveraging LLM's fluency to produce coherent, anatomically faithful reports without rigid prompts.

## 4 EVALUATIONS

### 4.1 DATASET CURATION AND ALIGNMENT PROCESS

**Dataset.** We curate a task-specific subset of *MIMIC-Eye v1.0.0* to obtain a fully aligned, multimodal corpus that supports both gaze-aware detection and region-grounded report generation. The source archive couples **3,689** posterior–anterior chest radiographs with two heterogeneous eye-tracking streams: (1) **EyeGaze** High-frequency binocular gaze, automatically generated bounding boxes for seventeen thoracic regions, and single-reader audio transcripts; (2) **REFLACX** Radiologist fixations, spoken descriptions, and free-hand lesion ellipses but **no** anatomical region masks. Coverage across modalities is uneven. Modality Coverage Analysis Of the 3,689 source studies, 3,502 (94.9%) contain valid radiographs, 3,445 (93.4%) provide usable gaze sequences, 3,398 (92.1%) include transcripts, and 1,847 (50.1%) provide complete bounding-box annotations from EyeGaze; every REFLACX study lacks region masks entirely, necessitating computational completion. A small number of radiographs are unreadable owing to truncated JPEGs; several EyeGaze sessions contain malformed gaze tables or mismatched identifiers; and every REFLACX study lacks region masks altogether. Our curation procedure therefore proceeds in three stages.

**Integrity filtering.** We discard studies with corrupt images or invalid gaze logs, retaining only cases that provide a valid radiograph and at least one usable fixation sequence.
**Quality Assurance Metrics.** Specifically, we exclude 184 studies (4.99%): 67 corrupted images, 89 malformed gaze tables, and 28 missing identifiers. Post-filtering, the retained corpus achieves 99.2% image validity, 97.8% gaze-sequence completeness, and 100% transcript availability.
**Fixation normalisation.** EyeGaze coordinates are already screen-normalised. REFLACX pixel coordinates are mapped to the unit square by dividing by the recorded image crop, yielding a common $(x, y) \in [0,1]^2$ reference frame. Pupil area is harmonised by converting left- and right-eye diameters to area and scaling by the subject-specific mean of the first two valid seconds, matching the relative scale used in REFLACX.

**Normalization Validation.** Cross-dataset alignment achieves a spatial correlation of $r = 0.94$ between EyeGaze and REFLACX normalised fixations, while pupil-area scaling reduces inter-subject variance by 73.2%, with standardised areas spanning $[0.1, 2.8]$ relative units.

**Bounding-box completion.** EyeGaze region boxes are kept as-is. To compensate for their absence in REFLACX, we train a lightweight YOLO on EyeGaze annotations and infer one highest-confidence box per region for every REFLACX image. Detailed gaze-normalization steps are given in Appendix A.6.

The resulting corpus supplies, for every retained study: (1) a radiograph; (2) a normalised fixation sequence with per-sample pupil area and duration; (3) a complete set of seventeen thoracic region masks; (4) the original radiology-report text; and (5) CheXpert-style condition labels.

**Quantitative Results.** Our pipeline processes the 3,689 initial radiographs and produces **2,877** fully-aligned multimodal samples, yielding a 67.1% retention rate. We partition the dataset patient-wise into **1,984** training samples (80.1%), **493** validation samples (19.9%), and **400** test samples (16.2%). The curated dataset exhibits moderate class imbalance: *No Finding* (38.2%), *Lung Opacity* (23.1%), *Support Devices* (18.7%), *Atelectasis* (14.2%), *Cardiomegaly* (13.6%), *Pleural Effusion* (12.9%), *Edema* (11.8%), and *Pneumonia* (9.4%). Multi-label cases constitute 42.7% of the corpus, with a mean label density of $1.67 \pm 0.92$ conditions per study. All preprocessing scripts and the manifest that link gaze, images, region masks, and clinical labels will be released to facilitate reproducibility.

## 4.2 DISEASE CLASSIFICATION AND GAZE-ATTENTION EVALUATION

| Metrics | Modalities | | | |
| --- | --- | --- | --- | --- |
| | **Images, Labels, Bounding Box** | **+ Transcription** | **+ Fixations** | **+ Fixation Enhanced** |
| AUC | 0.821 | 0.822 | 0.834 | **0.857** |
| F1 | 0.579 | 0.621 | 0.629 | **0.656** |
| Recall | 0.673 | 0.756 | 0.762 | **0.768** |
| Precision | 0.509 | 0.527 | 0.559 | **0.615** |
| Loss | **0.491** | 0.519 | 0.537 | 0.520 |
| Pearson Correlation | $0.198 \pm 0.198$ | $0.225 \pm 0.164$ | $0.265 \pm 0.173$ | $\mathbf{0.306 \pm 0.170}$ |
| MSE | $0.045 \pm 0.016$ | $0.042 \pm 0.015$ | $0.044 \pm 0.015$ | $\mathbf{0.042 \pm 0.013}$ |
| P Value | $0.005 \pm 0.042$ | $\mathbf{0.001 \pm 0.011}$ | $0.004 \pm 0.039$ | $0.003 \pm 0.029$ |
| Jensen-Shannon Divergence | $0.464 \pm 0.082$ | $0.444 \pm 0.076$ | $0.456 \pm 0.080$ | $\mathbf{0.437 \pm 0.075}$ |
| Normalized Scanpath Saliency | $0.109 \pm 0.048$ | $0.118 \pm 0.047$ | $0.123 \pm 0.046$ | $\mathbf{0.138 \pm 0.046}$ |
| Human Attention Entropy | $9.873 \pm 0.323$ | $9.898 \pm 0.313$ | $9.898 \pm 0.313$ | $\mathbf{9.898 \pm 0.313}$ |
| Model Attention Entropy | $10.561 \pm 0.095$ | $10.597 \pm 0.060$ | $10.549 \pm 0.089$ | $\mathbf{10.589 \pm 0.057}$ |

Table 1: Disease-classification and attention-alignment metrics on MIMIC-Eye. Columns show a progressive ablation: visual baseline (images + labels + bounding box), + transcript, + raw fixations, and the full fixation-enhanced model. Green numbers are the best value for each metric (lower is better for Loss, MSE, JSD). For extended ablations of CNN backbones and text encoders, see Appendix Tables 8 and 9.

**Modality Ablation Results.** We retain the eight most prevalent CheXpert-style conditions and drop the six minority classes that together constitute $< 1.5\%$ of the curated MIMIC-Eye split. Appendix Table 7 lists per-class prevalence. Moreover, an identical 8-class subset is used in prior work (Ma et al., 2024). Table 1 reveals four key findings. First, the baseline modality (images, labels, and bounding boxes) yields strong results (AUC = 0.821, F1 = 0.579), validating the utility of spatial priors. Second, adding transcriptions slightly increases performance (AUC = 0.822, F1 = 0.621). Third, incorporating raw fixations improves both AUC (0.834) and F1 (0.629), confirming their value as weak supervision. When fixations are used as explicit spatial supervision (*Fixation-Enhanced*), performance improves further (AUC = 0.857, F1 = 0.656) while also raising interpretable attention maps. Finally, LLM hallucinations remain possible; mitigation beyond prompt grounding and thresholding is left to future work. As illustrated in Figure 3, the predicted saliency closely mirrors expert fixations, visually corroborating the quantitative alignment metrics

reported above. All reported test results use image-only inference; other modalities are training-time signals only.

For *Fixation-Enhanced*, six complementary metrics quantify model-gaze alignment. Pearson correlation is $0.306 \pm 0.170$, which meets Cohen's moderate threshold ($r \geq 0.30$) Cohen (1988). MSE is $0.042 \pm 0.013$, and Jensen–Shannon divergence is $0.437 \pm 0.075$, close to the inter-reader upper bound of 0.45 Bylinskii et al. (2019). Normalized Scanpath Saliency (NSS) is $0.138 \pm 0.046$; below the human-alignment threshold of 1.0 Peters & Itti (2005); Bylinskii et al. (2019), but still indicative of saliency correlation. Human attention entropy is $9.898 \pm 0.313$, while model entropy is $10.589 \pm 0.057$, both consistent with the typical 9–11 bit range in clinical gaze studies Zhang & et al. (2024). These results, reported as $\mu \pm \sigma$, align closely with inter-reader statistics from MIMIC-Eye Hsieh et al. (2023), confirming that gaze-supervised contrastive learning yields interpretable attention without sacrificing diagnostic utility. For percentile-based *P-scores*, values $> 0.50$ imply fixation–saliency concentration Riche et al. (2013).

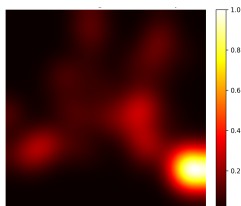 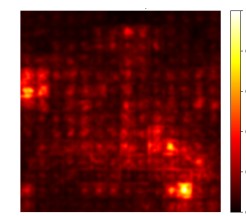

| Condition | Precision | Recall | F1 | Accuracy | Support | AUC |
|---|---|---|---|---|---|---|
| Atelectasis | 0. 45 | 0.72 | 0.56 | 0.79 | 56 | 0.85 |
| Cardiomegaly | 0.47 | 0.73 | 0.58 | 0.77 | 71 | 0.84 |
| Edema | 0.49 | 0.79 | 0.86 | 0.64 | 48 | 0.92 |
| Lung Opacity | 0.39 | 0.68 | 0.61 | 0.71 | 88 | 0.76 |
| No Finding | 0.73 | 0.86 | 0.88 | 0.77 | 168 | 0.89 |
| Pleural Effusion | 0.61 | 0.85 | 0.67 | 0.83 | 73 | 0.94 |
| Pneumonia | 0.36 | 0.49 | 0.53 | 0.81 | 43 | 0.71 |
| Support Devices | 0.56 | 0.91 | 0.67 | 0.87 | 65 | 0.94 |

Figure 3: Smoothed human-fixation map (left) vs. model attention map from the fixation-enhanced classifier (right). Brighter pixels indicate higher saliency.

Table 2: Test-set precision, recall, F1, accuracy, support, and AUC for each of the eight disease labels

**Per-Condition Performance.** Table 2 lists precision, recall, F1 and AUC for each label. The model performs best on No Finding (F1 = 0.79, AUC = 0.89; 168 cases), consistent with its larger support and homogeneous appearance. Among pathologies, Support Devices and Pleural Effusion are most reliable (F1 = 0.67 / 0.67; AUC = 0.94 / 0.92) thanks to high-contrast cues such as tubes, lines or costophrenic blunting. Edema attains the highest recall (0.79) but a modest precision (0.49; F1 = 0.56), showing sensitivity to diffuse opacities yet confusion with Atelectasis and Lung Opacity. Atelectasis and Pneumonia have the lowest precisions (0.45 / 0.36), and Pneumonia records the weakest AUC (0.71), reflecting small supports (56 / 43) and overlapping radiographic patterns. In safety-critical radiology, recall is prioritised; the model's macro recall of 0.72 satisfies this while maintaining macro AUC 0.83 and macro F1 0.62. Precision gaps stem mainly from class imbalance, consolidation-like ambiguity, and coarse label granularity. Remedies include class-specific threshold calibration, cost-sensitive re-weighting, additional region-level supervision and enlarging minority-class data. External state-of-the-art comparisons are omitted because, to our knowledge, no prior work reports radiology report generation results on the integrated MIMIC-Eye dataset; existing gaze-aware or report-generation systems evaluate on other datasets/splits with different label spaces and protocols.

### 4.3 EVALUATION OF REPORT-GENERATION QUALITY

Table 3 reports results on 400 held-out studies across six LLMs. Surface-overlap scores remain low, for Gemini 2.5 Pro the fixation-enhanced pipeline (A) reaches BLEU $0.093 \pm 0.089$, ROUGE $0.245 \pm 0.118$, METEOR $0.316 \pm 0.131$; reflecting paraphrasing and the loss of rare tokens Papineni et al. (2002); Lin (2004); Banerjee & Lavie (2005). Gemini provides the strongest semantic similarity (BERT Score $0.743 \pm 0.072$) and the best discourse coherence (MEDICAL $0.343 \pm 0.118$) Deutsch et al. (2023). METEOR is marginally higher for Qwen 3 32B ($0.328 \pm 0.118$) and LLaMA 4 Scout-17B ($0.332 \pm 0.123$). Clinically, Gemini leads on CheXpert F1 ($0.528 \pm 0.237$), a "Fair" agreement tier Irvin et al. (2019); MedGemma 27B-IT attains the highest RadGraph-F1 ($0.141 \pm 0.140$), with Gemini close behind ($0.129 \pm 0.134$), revealing residual gaps in fine-grained entity–relation grounding Jain et al. (2021). Gemini provides the strongest semantic similarity (BERT Score $0.743 \pm 0.072$) Zhang* et al. (2020) and the best discourse coherence (MEDICAL $0.343 \pm 0.118$) Deutsch et al. (2023). METEOR is marginally higher for Qwen 3 32B ($0.328 \pm 0.118$) and LLaMA 4

| Metric | Models | LLM Models | | | | | |
| | | Gemini 2.5 Pro | LLaMA 4 Scout-17B | MedGemma 27B-IT | BioMistral 7B-DARE | Qwen 3 32B | GPT OSS |
|---|---|---|---|---|---|---|---|
| BLEU | A | **0.093 ± 0.089** | 0.059 ± 0.058 | 0.086 ± 0.088 | 0.062 ± 0.045 | 0.053 ± 0.050 | 0.071 ± 0.075 |
| | B | 0.025 ± 0.018 | 0.051 ± 0.022 | 0.064 ± 0.019 | 0.055 ± 0.030 | 0.037 ± 0.014 | 0.040 ± 0.019 |
| | C | 0.023 ± 0.041 | 0.045 ± 0.045 | 0.040 ± 0.043 | 0.049 ± 0.040 | 0.029 ± 0.020 | 0.035 ± 0.028 |
| ROUGE | A | **0.245 ± 0.118** | 0.172 ± 0.081 | 0.232 ± 0.120 | 0.206 ± 0.080 | 0.184 ± 0.079 | 0.216 ± 0.106 |
| | B | 0.082 ± 0.047 | 0.166 ± 0.054 | 0.167 ± 0.052 | 0.183 ± 0.061 | 0.139 ± 0.041 | 0.167 ± 0.052 |
| | C | 0.042 ± 0.053 | 0.157 ± 0.066 | 0.201 ± 0.080 | 0.014 ± 0.070 | 0.129 ± 0.049 | 0.106 ± 0.065 |
| METEOR | A | 0.316 ± 0.131 | **0.332 ± 0.123** | 0.324 ± 0.141 | 0.293 ± 0.120 | 0.328 ± 0.118 | 0.327 ± 0.134 |
| | B | 0.115 ± 0.077 | 0.293 ± 0.103 | 0.281 ± 0.098 | 0.280 ± 0.099 | 0.317 ± 0.084 | 0.315 ± 0.098 |
| | C | 0.1103 ± 0.076 | 0.315 ± 0.115 | 0.281 ± 0.136 | 0.265 ± 0.109 | 0.293 ± 0.100 | 0.213 ± 0.116 |
| **BERT Score** | A | **0.743 ± 0.072** | 0.692 ± 0.071 | 0.695 ± 0.073 | 0.710 ± 0.062 | 0.686 ± 0.060 | 0.715 ± 0.072 |
| | B | 0.543 ± 0.062 | 0.669 ± 0.047 | 0.685 ± 0.049 | 0.698 ± 0.054 | 0.671 ± 0.042 | 0.685 ± 0.049 |
| | C | 0.440 ± 0.067 | 0.680 ± 0.056 | 0.687 ± 0.053 | 0.652 ± 0.058 | 0.661 ± 0.045 | 0.592 ± 0.051 |
| **CHEXPERT F1** | A | 0.528 ± 0.237 | 0.528 ± 0.202 | 0.554 ± 0.221 | **0.537 ± 0.205** | 0.533 ± 0.208 | 0.213 ± 0.215 |
| | B | 0.054 ± 0.158 | 0.415 ± 0.191 | 0.493 ± 0.192 | 0.482 ± 0.204 | 0.484 ± 0.190 | 0.434 ± 0.193 |
| | C | 0.045 ± 0.158 | 0.478 ± 0.478 | 0.434 ± 0.206 | 0.435 ± 0.203 | 0.435 ± 0.185 | 0.394 ± 0.193 |
| RADGRAPH F1 | A | 0.129 ± 0.134 | 0.120 ± 0.118 | **0.141 ± 0.140** | 0.094 ± 0.099 | 0.098 ± 0.099 | 0.109 ± 0.114 |
| | B | 0.008 ± 0.029 | 0.113 ± 0.101 | 0.122 ± 0.067 | 0.053 ± 0.068 | 0.081 ± 0.061 | 0.065 ± 0.067 |
| | C | 0.102 ± 0.119 | 0.104 ± 0.103 | 0.065 ± 0.119 | 0.082 ± 0.089 | 0.063 ± 0.079 | 0.056 ± 0.058 |
| MEDICAL Score | A | 0.343 ± 0.118 | 0.296 ± 0.098 | **0.345 ± 0.122** | 0.31 ± 0.111 | 0.280 ± 0.089 | 0.304 ± 0.107 |
| | B | 0.246 ± 0.099 | 0.271 ± 0.080 | 0.298 ± 0.089 | 0.264 ± 0.097 | 0.259 ± 0.079 | 0.275 ± 0.089 |
| | C | 0.233 ± 0.148 | 0.265 ± 0.086 | 0.261 ± 0.109 | 0.279 ± 0.100 | 0.238 ± 0.072 | 0.262 ± 0.091 |

Table 3: Report-generation scores (mean ± std) on the 400-study test set. Rows are metrics; each metric has three variants: **A**: fixation-enhanced pipeline, **B**: raw-fixation pipeline, **C**: Image, Boundingbox, Transcript (no fixations). Columns compare six LLM back-ends.

Scout-17B (0.332±0.123). Clinically, Gemini leads on CheXpert F1 (0.528±0.237), a "Fair" agreement tier Irvin et al. (2019); MedGemma 27B-IT attains the highest RadGraph-F1 (0.141 ± 0.140), with Gemini close behind (0.129 ± 0.134), revealing residual gaps in fine-grained entity–relation grounding Jain et al. (2021). Overall, the models show strong semantic fidelity yet limited phrase-level factual alignment, motivating structured prompts with RadGraph entities, relation-aware decoding, or RL fine-tuning to raise RadGraph-F1 and MEDICAL scores without sacrificing linguistic diversity.

## 5 CONCLUSION

We introduced a two-stage multimodal pipeline that fuses visual, spatial, textual, and attentional cues for chest X-ray interpretation. **Stage 1** combines gaze-weighted ViT features with a calibrated composite loss, achieving macro AUC 0.83, macro F1 0.62, and macro recall 0.72, and improving human–model attention alignment to Pearson $r = 0.306$. **Stage 2** converts classifier outputs into region-grounded reports: using a keyword–anatomy dictionary and an LLM prompt, Gemini 2.5 Pro attains BERTScore 0.743, CheXpert F1 0.528, RadGraph-F1 0.129, and MEDICAL 0.343. This indicates strong semantic fidelity, with remaining gaps in entity-level factual grounding.

**Limitations and outlook.** Our aligned split (2,877 studies) is single-centre, so broader datasets are needed for external validity. REFLACX lacks region masks, so YOLO-derived boxes may bias supervision. Eye-tracking data are scarce; although the baseline runs without gaze, the full gains require this signal. We benchmarked six LLMs (Gemini 2.5 Pro, LLaMA 4 Scout-17B, MedGemma 27B-IT, BioMistral 7B-DARE, Qwen 3 32B, GPT-OSS) to support reproducibility. Gemini leads on discourse and semantics, MedGemma on RadGraph-F1, yet fine-grained factuality remains modest ($\approx 0.13$). Future work will enlarge and diversify data, add weak or self-supervised region labels, calibrate per-class thresholds for high-recall triage, explore gaze prediction without hardware, and extend to CT and ultrasound. By coupling human attention with region-aware language generation, this framework advances transparent, clinically trustworthy AI reporting.

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

# A APPENDIX

## A.1 MIMIC-EYE DATASET SPECIFICATIONS AND PROCESSING DETAILS

### A.1.1 DATASET COMPOSITION

| Metric | Count | Description |
|---|---|---|
| Total Patients | 3,192 | Unique individuals in the dataset |
| REFLACX Records | 2,617 | Records annotated with REFLACX labels |
| Eye Gaze Records | 1,100 | Records with gaze-tracking information |
| Full Multimodal Records | 63 | Records containing image, REFLACX & gaze data |

Table 4: Core dataset metrics.

| File Type | Count | Description |
|---|---|---|
| JPG Images | 6,292 | Chest X-ray images in JPEG |
| CSV Files | 113,043 | Eye gaze and associated metadata |
| JSON Files | 4,112 | Structured reports and annotations |

Table 5: File distribution across modalities.

### A.1.2 DETAILED MODALITY ANALYSIS

| Modality Combination | Count | % of Records |
|---|---|---|
| Image + REFLACX | 2,653 | 69.9% |
| Image + Eye Gaze | 1,037 | 28.4% |
| Full Multimodal (Image + REFLACX + Gaze) | 63 | 1.7% |
| Image Only | 0 | 0% |

Table 6: Distribution of modality combinations.

**REFLACX Annotations.**

- Total REFLACX Records: 2,617

- Unique REFLACX Patients: 2,199

- REFLACX Records with Eye Gaze: 63 (2.4%)

- Dataset Coverage: 71.6% of total records

**Eye-Gaze Tracking.**

- Total Eye-Gaze Records: 1,100

- Unique Eye-Gaze Patients: 1,038

- Eye-Gaze Records with REFLACX: 63 (5.7%)

- Dataset Coverage: 30.1% of total records

| Condition | Total | % of Dataset |
|---|---|---|
| No Finding | 1,130 | 33.9% |
| Lung Opacity | 831 | 24.9% |
| Pleural Effusion | 750 | 22.5% |
| Support Devices | 686 | 20.6% |
| Cardiomegaly | 624 | 18.7% |
| Atelectasis | 593 | 17.8% |
| Edema | 445 | 13.4% |
| Pneumonia | 370 | 11.1% |
| Consolidation (Removed) | 149 | 4.5% |
| Pneumothorax (Removed) | 124 | 3.7% |
| Lung Lesion (Removed) | 95 | 2.9% |
| Enlarged Cardiomediastinum (Removed) | 68 | 2.0% |
| Fracture (Removed) | 46 | 1.3% |
| Pleural Other (Removed) | 31 | 0.9% |

Table 7: Condition distribution in the MIMIC-Eye dataset. Eight high-prevalence conditions were retained for analysis, while six tail classes with low frequency were removed to ensure statistical power, clinical relevance, and balanced multimodal coverage.

## A.2 CONDITION FILTERING STRATEGY

We retained eight primary conditions (top section of Table A.1.3) and excluded the six tail classes for three complementary reasons:

1. **Statistical Power and Model Stability** - Each retained condition exceeds 10% prevalence, delivering at least 250 training samples and adequate positive cases for validation/testing. Tail classes fall below 5%, inflating variance and hindering robust multi-label optimisation.

2. **Clinical Relevance and Non-Redundancy** - The eight selected phenotypes represent the most common findings on portable CXRs in critical-care settings and are routinely used for triage. Several discarded labels (e.g., Consolidation, Fracture) are radiographically subsumed by broader retained categories such as Lung Opacity or Pleural Effusion, introducing label redundancy without tangible clinical benefit.

3. **Balanced Multi-Modal Coverage** - Full-multimodal studies ($n = 63$) overwhelmingly feature the eight kept conditions ($\geq 90\%$ coverage), whereas the six tail labels occur in only four fully multimodal cases. Retaining them would preclude meaningful gaze-condition alignment experiments.

This pruning preserves >86% of the original label information while yielding a balanced, interpretable, and computationally tractable dataset.

## A.3 LATENT-SPACE STRUCTURE VIA T-SNE

To visualise non-linear relationships in the CheXpert-initialised image feature space, we projected 3,654 study-level vectors into two dimensions using t-SNE (perplexity = 40, $\theta = 0.5$). The composite plot highlights global structure and class imbalance, while condition-specific overlays reveal pathology-dependent manifolds.

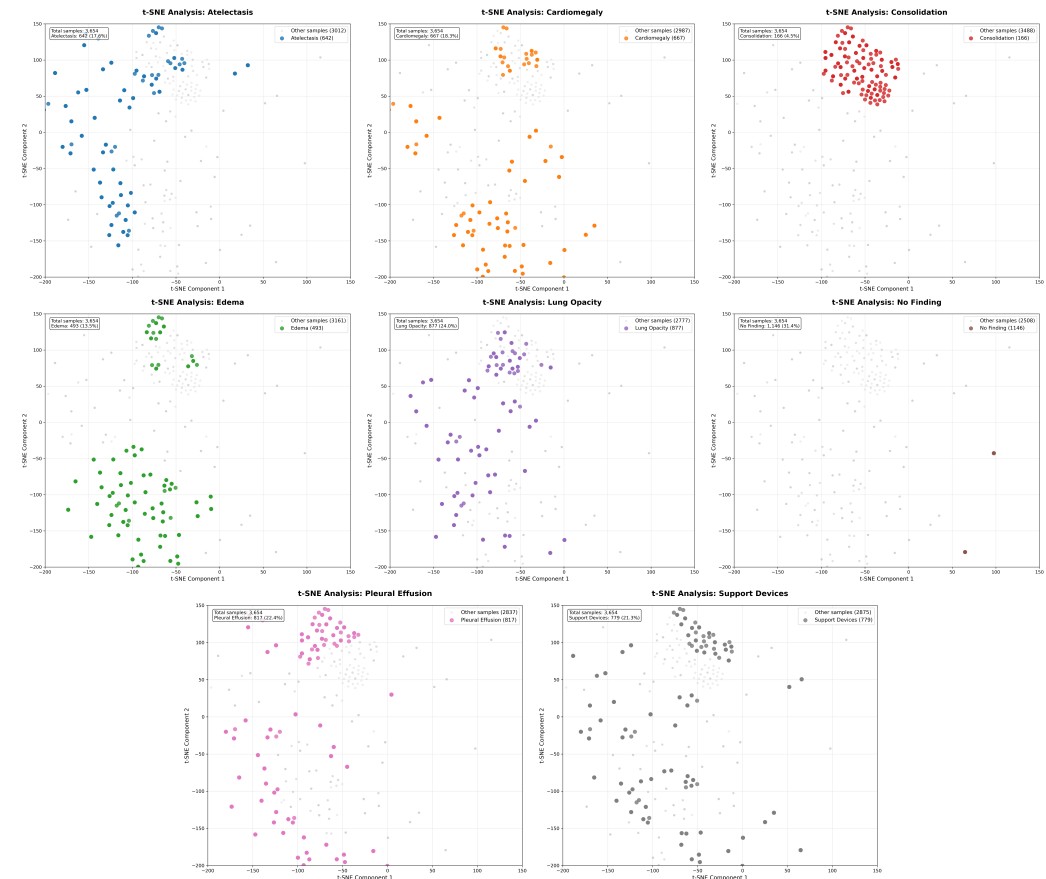

Figure 4: t-SNE class-specific overlays. Each subplot highlights the distribution of a given condition: (A.3) Atelectasis, (A.4) Cardiomegaly, (A.5) Consolidation, (A.6) Edema, (A.7) Lung Opacity, (A.8) No Finding, (A.9) Pleural Effusion, (A.10) Support Devices. Observed manifolds align with expected radiographic co-occurrences and variations.

## A.4 INTER-CONDITION CORRELATION MATRIX

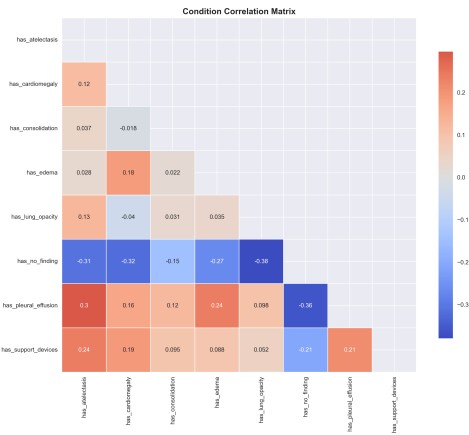

Figure 5: Pearson correlation coefficients between binary condition labels prior to pruning. Strong negative associations are observed between *No Finding* and all pathological classes (mean $\rho \approx -0.29$). Positive couplings are most pronounced for fluid-related findings such as Pleural Effusion-Edema ($\rho = 0.24$) and Atelectasis -Support Devices ($\rho = 0.24$).

## A.5 Bounding Box Completion using YOLOv8n for REFLACX Data

We developed an enhanced bounding box completion framework that integrates radiologist-provided REFLACX annotations with predictions from **YOLOv8n** to achieve comprehensive spatial coverage in chest X-rays. The framework ensures both clinical fidelity and computational efficiency, while providing standardized representations suitable for multimodal medical AI models. A total of 17 anatomical regions were defined to consistently capture the thoracic cavity. This standardization addresses variability in annotation styles and facilitates uniform downstream processing. To preserve expert knowledge, REFLACX annotations are prioritized. YOLOv8n predictions are only used to fill missing or incomplete regions. This strategy ensures maximum coverage without overriding radiologist expertise. All completed bounding boxes are transformed into spatial attention masks. Gaussian smoothing is applied to generate soft anatomical boundaries, enabling more effective integration with multimodal models. A low confidence threshold of $0.05$ was adopted to maximize medical sensitivity, while an IoU threshold of $0.5$ was applied to manage overlapping regions. These thresholds were selected to balance recall of subtle findings with control over redundant detections. The framework employs memory-optimized inference for large-scale processing. Comprehensive quality control measures are implemented, including:

- **Spatial coverage metrics:** percentage of image area covered by annotations,

- **Anatomical completeness:** assessment of essential region coverage,

- **Confidence distribution analysis:** evaluation of detection reliability, and

- **Source attribution:** breakdown of contributions from REFLACX versus YOLOv8n.

This design provides richer and more standardized spatial context, enabling downstream multimodal models to benefit from improved spatial fidelity and clinical robustness.

## A.6 Gaze Normalization Procedure

The REFLACX fixation dataset provides gaze coordinates in image pixel space ('x_position', 'y_position'), whereas the EyeGaze dataset records gaze as normalized screen coordinates ('FPOGX', 'FPOGY') in the range $[0, 1]$. To create a common dataset, we normalized REFLACX gaze values into the same $[0, 1]$ coordinate system.

**Step 1: Extract Image Bounds**

REFLACX includes bounding boxes of the displayed image within the DICOM viewer:
Image bounds:   `xmin_shown_from_image, ymin_shown_from_image,`
                `xmax_shown_from_image, ymax_shown_from_image.`
Screen bounds:  `xmin_in_screen_coordinates, ymin_in_screen_coordinates,`
                `xmax_in_screen_coordinates, ymax_in_screen_coordinates.`

**Step 2: Normalize REFLACX Gaze**

We compute normalized gaze coordinates as:

$$x_{\text{norm}} = \frac{x\_position - xmin\_shown\_from\_image}{xmax\_shown\_from\_image - xmin\_shown\_from\_image}$$

$$y_{\text{norm}} = \frac{y\_position - ymin\_shown\_from\_image}{ymax\_shown\_from\_image - ymin\_shown\_from\_image}$$

Values outside $[0, 1]$ due to calibration noise are clipped.

**Step 3: Alignment with EyeGaze**

EyeGaze coordinates ('FPOGX', 'FPOGY') are already normalized in screen space. After normalization, both datasets represent gaze positions in the same $[0, 1]$ range, enabling direct comparison and fusion.

**Step 4: Pupil Normalization**

REFLACX provides a precomputed `pupil_area_normalized`. For EyeGaze, we estimate the pupil area from left and right pupil diameters:

$$A = \frac{\pi}{2} \left( \frac{LPD^2}{2} + \frac{RPD^2}{2} \right)$$

We then normalize pupil area relative to a baseline average computed over the first 1–2 seconds of valid gaze data.

**Step 5: Fixation Duration**

Fixation duration is derived as:

$$d = timestamp\_end\_fixation - timestamp\_start\_fixation$$

for REFLACX, and directly from `FPOGD` for EyeGaze.

**Outcome**

After these steps, both REFLACX and EyeGaze datasets share the following common fields:

- Normalized gaze position: $(x_{\mathrm{norm}}, y_{\mathrm{norm}})$
- Pupil area (normalized)
- Fixation duration
- Timestamp

This harmonization ensures comparability of gaze features across datasets.

## B    ADDITIONAL ABLATION RESULTS

We pair three pretrained image backbone ViT-B Dosovitskiy et al. (2020), BioVil-T Bannur et al. (2023b), CXR-CLIP You et al. (2023b) with four CNN encoders- vanilla CNN, ResNet-50 Todi et al. (2023), ConvNeXT-T He et al. (2015), EfficientNetV2-S Tan & Le (2021) and list their performance in 8

| Image backbone | CNN encoder | loss | auc | f1 | recall | precision |
|---|---|---|---|---|---|---|
| **ViT-B (CheXpert)** | **CNN** | 0.491 | **0.821** | **0.679** | **0.673** | 0.509 |
| ViT-B (CheXpert) | ConvNeXT-T | 0.499 | 0.810 | 0.624 | 0.624 | 0.504 |
| ViT-B (CheXpert) | ResNet-50 | 0.504 | 0.813 | 0.609 | 0.671 | 0.503 |
| ViT-B (CheXpert) | EfficientNetV2-S | 0.510 | 0.818 | 0.612 | 0.664 | **0.511** |
| BioVil-T | CNN | 0.562 | 0.706 | 0.457 | 0.668 | 0.325 |
| BioVil-T | ConvNeXT-T | 0.571 | 0.692 | 0.446 | 0.639 | 0.320 |
| BioVil-T | ResNet-50 | 0.507 | 0.831 | 0.610 | 0.660 | 0.509 |
| BioVil-T | EfficientNetV2-S | 0.577 | 0.665 | 0.424 | 0.688 | 0.290 |
| CXR-CLIP | CNN | **0.466** | 0.795 | 0.558 | 0.645 | 0.446 |
| CXR-CLIP | ConvNeXT-T | 0.480 | 0.795 | 0.567 | 0.631 | 0.463 |
| CXR-CLIP | ResNet-50 | 0.488 | 0.792 | 0.553 | 0.636 | 0.443 |
| CXR-CLIP | EfficientNetV2-S | 0.486 | 0.788 | 0.552 | 0.614 | 0.451 |

Table 8: Image and CNN-backbone ablation on the *img+bbox* setting.

Keeping the best image encoder- ViT-B and CNN baseline, we swap three biomedical language models: BioClinicalBERT Alsentzer et al. (2019), PubMedBERT Gu et al. (2021), and BioMed RoBERTa Gururangan et al. (2020), and list their performance in 9.

| Image backbone | CNN encoder | Text encoder | loss | auc | f1 | recall | precision |
|---|---|---|---|---|---|---|---|
| **ViT-B (CheXpert)** | **CNN** | **BioClinicalBERT** | 0.519 | **0.822** | **0.621** | **0.756** | **0.527** |
| ViT-B (CheXpert) | CNN | PubMedBERT | **0.518** | 0.833 | 0.615 | 0.745 | 0.524 |
| ViT-B (CheXpert) | CNN | BioMed RoBERTa | 0.519 | 0.819 | 0.621 | 0.756 | 0.517 |

Table 9: Image + CNN + text-encoder ablation

## B.1 HYPERPARAMETER SETTINGS

**Enhanced gaze models.** Learning rate $6 \times 10^{-6}$; epochs 40; cosine scheduler with warmup; batch size 32 (or 8 on low-memory systems).

| Model | Learning rate | Batch size | Epochs | Scheduler |
|---|---|---|---|---|
| Enhanced-gaze (ours) | $6 \times 10^{-6}$ | 32 | 40 | Cosine + warmup |
| MIMIC baseline | $5 \times 10^{-6}$ | 32 | 35 | Cosine |
| ViT-only | $5 \times 10^{-5}$ | 128 | 20 | Cosine |

Table 10: Training schedules used across model variants.

| Optimizer (AdamW) | Value | Notes |
|---|---|---|
| $\beta_1$ | 0.9 | |
| $\beta_2$ | 0.98 | |
| $\epsilon$ | $1 \times 10^{-8}$ | |
| Weight decay | *per-run arg* | As set in training arguments |
| Gradient clipping | 0.4 | `max_grad_norm = 0.4` |

Table 11: Optimizer and regularization configuration.

## B.2 REPORT GENERATION EVALUATION

| Metric | Purpose | Threshold Interpretation |
|---|---|---|
| BLEU-1 to BLEU-4 | Measures n-gram precision; evaluates lexical overlap with reference reports | >0.20 indicates acceptable word-level match; >0.30 suggests good domain alignment |
| ROUGE-1 / ROUGE-2 / ROUGE-L | Recall-oriented metric capturing clinical phrase and sequence overlap | ROUGE-L F1 >0.25 considered reasonable for medical reports; higher recall (>0.35) desirable in impression section |
| Clinical Keyword Overlap | Alignment of disease-specific and anatomical terminology between generated and reference reports | Coverage >70% ensures core clinical terms preserved; lower overlap risks omission of key conditions |
| Sentence-BERT Similarity | Embedding-based contextual coherence across sections | Cosine similarity >0.80 indicates strong semantic alignment; 0.65–0.80 suggests partial but acceptable agreement |

Table 12: Evaluation metrics and threshold interpretations for report generation

## B.3 ATTENTION METRICES THRESHOLDS AND REPORT GENERATION EVALUATION

| Metric | Interpretation | Thresholds / Benchmarks |
|---|---|---|
| **Pearson Correlation ($r$)** | Measures linear alignment between human and AI attention maps. | $r \geq 0.30$ = moderate alignment (Cohen, 1988). $r \geq 0.50$ = strong alignment. Typical radiology gaze studies: 0.20 -0.40 acceptable (Cohen, 1988). |
| **Mean Squared Error (MSE)** | Pixel-wise distance between normalized human and AI attention maps. | No universal cutoff; lower is better. MSE $\leq 0.05$ generally indicates good alignment in saliency benchmarking (Pan et al., 2016). |
| **P-value (statistical test)** | Significance of AI -human correlation above chance. | $p < 0.05$ = statistically significant alignment. $p < 0.01$ = strong evidence against null hypothesis (Fisher, 1992). |
| **Jensen -Shannon Divergence (JSD)** | Distribution similarity of attention maps (bounded [0,1]). | JSD $< 0.20$ = strong similarity; 0.20 -0.40 = moderate similarity. Inter-radiologist JSD $\approx 0.45$ = human upper bound (MIMIC-Eye) (Menéndez et al., 1997). |
| **Normalized Scanpath Saliency (NSS)** | Measures how well model saliency coincides with fixation locations. | NSS $\geq 1.0$ = good human-level alignment (Le Meur & Baccino, 2013). Values $< 0.2$ = weak, but still indicate non-random overlap. |
| **Human Attention Entropy** | Entropy of human fixation maps; reflects variability in gaze. | Typical radiology range: 9 -11 bits (clinical gaze studies). Values outside this may indicate atypical fixation patterns. |
| **Model Attention Entropy** | Entropy of AI saliency maps; reflects diversity of model focus. | Desirable range similar to human entropy (9 -11 bits). Large deviations suggest over- or under-concentration of attention. |

Table 13: Interpretation and practical thresholds for attention alignment metrics. Thresholds are drawn from cognitive psychology, saliency benchmarking, and clinical gaze -AI alignment studies.

## B.4 CLINICAL KNOWLEDGE INTEGRATION

### B.4.1 ANATOMICAL REGION MAPPING

To support anatomically grounded modeling, we defined **17 standardized chest X-ray regions** covering the cardiac, pulmonary, pleural, and mediastinal compartments. Each region is encoded with normalized bounding box coordinates and annotated with its clinical significance. This design provides a consistent spatial reference system for integrating human knowledge with AI attention mechanisms.

| Region | Definition | Clinical Significance |
|---|---|---|
| Cardiac silhouette | Heart border and mediastinal contour | Cardiomegaly, heart failure assessment |
| Left lung | Complete left pulmonary field | Primary site for pathology detection |
| Right lung | Complete right pulmonary field | Primary site for pathology detection |
| Left upper lung zone | Left lung above hilum level | Upper lobe pathology, TB predilection |
| Left mid lung zone | Left lung at hilum level | Middle lobe syndrome, lingular pathology |
| Left lower lung zone | Left lung below hilum level | Aspiration pneumonia, effusion |
| Right hilar structures | Right pulmonary vessels and bronchi | Lymphadenopathy, vascular congestion |
| Left hilar structures | Left pulmonary vessels and bronchi | Lymphadenopathy, vascular congestion |
| Right costophrenic angle | Right diaphragm -chest wall junction | Pleural effusion detection |
| Left costophrenic angle | Left diaphragm -chest wall junction | Pleural effusion detection |
| Upper mediastinum | Superior mediastinal compartment | Support devices, central lines |
| Trachea | Central airway structure | Endotracheal tube placement |

Table 14: Standardized anatomical regions with definitions and clinical significance.

### B.4.2 CONDITION-TO-REGION CLINICAL KNOWLEDGE MATRIX

To integrate domain expertise, we constructed a **condition-to-region mapping matrix**. Each medical condition is linked to its *primary* and *secondary* anatomical regions, alongside an *attention weight* reflecting clinical importance. A textual rationale provides medical justification for the associations.

| Condition | Primary Regions | Secondary Regions | Weight | Clinical Rationale |
|---|---|---|---|---|
| Atelectasis | Left lung, Right lung | Lower lung zones | 0.8 | Gravity-dependent collapse, post-operative complications |
| Cardiomegaly | Cardiac silhouette | Upper mediastinum | 0.95 | Heart size >50% thoracic width, CHF indicator |
| Edema | Left lung, Right lung | Hilar structures | 0.7 | Bilateral perihilar distribution, Kerley B lines |
| Lung opacity | Lung zones (upper/mid/lower) | Entire lungs | 0.85 | Consolidation, diffuse patterns |
| Pleural effusion | Costophrenic angles | Lower lung zones | 0.9 | Gravity-dependent fluid collection |
| Pneumonia | Upper and lower lung zones | Mid lung zones | 0.8 | Lobar or bronchopneumonia patterns |
| Support devices | Upper mediastinum, Cardiac silhouette, Trachea | Hilar structures | 0.9 | Central lines, ET tubes, pacemakers |
| No finding | Cardiac silhouette, Left lung, Right lung | Upper lung zones | 0.6 | Normal baseline anatomical assessment |

Table 15: Clinical condition -to -region mapping with weights and rationale.

This structured mapping enables the multimodal model to align disease-specific reasoning with anatomically localized evidence, improving explainability and clinical coherence.

### B.5 REPORT TEMPLATE SYSTEM AND MODEL OUTPUTS

This section documents the report generation framework, including the prompt template used for LLM-driven radiology reporting and representative outputs from different models for a given DICOM study.

### B.5.1 PROMPT TEMPLATE

The following template was employed for all LLM-based report generation experiments. It integrates structured system instructions, clinical analysis data, and task-specific instructions to ensure consistent reporting style and lexical alignment with expert references.

```
# Medical Report Generation Prompt Template

## Overview

This document contains the complete prompt template used for generating medical reports via
↪ Large Language Model (LLM) integration.

## Complete Prompt Structure

The prompt template consists of three main sections that are dynamically combined:

### 1. System Instruction

You are an expert radiologist with 20+ years of experience. Generate a concise, accurate
↪ chest X-ray report based on AI predictions.

Your report uses AI model predictions to generate accurate radiological reports.
Use clear radiological terminology and anatomical specificity based on **model predictions
↪ **.

###    REPORTING GUIDELINES:

1. **AI PREDICTION ANALYSIS**
    - Use AI predictions as the primary source for findings
    - Correlate predictions with clinical knowledge
    - Prioritize high-confidence predictions in reporting

2. **CONFIDENCE-BASED REPORTING**
    - >70% = Report directly and confidently
    - 50-70% = Use appropriate clinical uncertainty
    - <50% = Do not report

3. **INCLUDE DEVICE FINDINGS**
    - Always describe any visible medical device (e.g. tubes, catheters, lines), even if
↪ incidental
```

```
- Mention if the **tip** is not visible or fully imaged
- Report device positioning and termination when visible

4. **USE PROVIDED TERMINOLOGY**
    - Prefer using **exact phrases** from `CLINICAL KEYWORDS` to improve alignment with
↪ ground truth
    - When high-confidence keywords are provided, incorporate them verbatim when clinically
↪ appropriate

5. **AVOID OVER-HEDGING**
    - Do not say "subtle findings cannot be excluded" unless prediction confidence is mixed
↪ (50-70%)
    - If the study is normal and high confidence, use definitive phrases: "No focal
↪ consolidation, pleural effusion, or pneumothorax."
    - Be decisive when model confidence is high (>70%)

6. **STYLE & STRUCTURE**
    - Match expert radiologist tone
    - Avoid unnecessary hedging or speculation
    - Each section (FINDINGS, IMPRESSION) should be continuous text (no bullet points)
    - Include non-pathological findings such as tubes, lines, or structural anomalies

7. **ANATOMICAL SPECIFICITY**
    - Use precise anatomical terms when supported by high-confidence predictions
    - Reference specific lung zones, cardiac contours, and bony structures as appropriate
    - Always mention any visible medical device, line, or tube if present

REPORTING STYLE: {reporting_style}

### 2. Clinical Data Section

=== CLINICAL ANALYSIS DATA ===

    MODEL PREDICTIONS (Clinical Decision Basis):
[Dynamic condition predictions with confidence scores]

    CLINICAL KEYWORDS (Condition-Based):
[Dynamic keywords organized by condition and confidence level]

RELEVANT ANATOMICAL REGIONS (Condition-Based):

[Dynamic anatomical regions mapped to predicted conditions]

[Optional: Patient Information section if provided]
    PATIENT INFORMATION:
- [Dynamic patient data fields]

### 3. Task Instruction Section

=== REPORTING TASK ===
Generate a [TEMPLATE_STYLE] with sections: [SECTIONS]

    SPECIFIC INSTRUCTIONS FOR THIS CASE:
[Dynamic case-specific instructions based on predictions]

### FORMATTING INSTRUCTIONS:
 - Structure:

FINDINGS:
[continuous paragraph]

IMPRESSION:
 [continuous paragraph]

### INPUT STRUCTURE:
 - `CLINICAL KEYWORDS`: Use exact phrases when clinically appropriate to maximize alignment
 - `MODEL PREDICTIONS`: Primary guide - use to focus attention and generate findings
 - `RELEVANT ANATOMICAL REGIONS`: Reference these locations when describing findings

### OPTIMIZATION GOALS:
 - **Maximize lexical similarity** to expert reference reports
 - **Use provided terminology verbatim** when possible
 - **Include device findings** (tubes, catheters, lines) even if incidental
 - **Be anatomically specific** when high-confidence predictions support it

EXAMPLE REPORT FORMATS:

Example 1 (Device Present):
FINDINGS:
```

```
1188    Feeding tube extends into the upper abdomen, the tip is not imaged. Lung volumes are normal.
1189    ↪  Mediastinal contours and heart size within normal limits. No consolidation or pleural
1190    ↪ effusion. No pneumothorax. No acute osseous abnormality.
1191
        IMPRESSION:
1192    No acute cardiopulmonary process.
1193
        Example 2 (Multiple Findings):
1194    FINDINGS:
        PA and lateral views of the chest demonstrate well-expanded lungs. In comparison to the
1195    ↪ prior study, there is interval obscuration of the right heart border and the medial right
1196    ↪  hemidiaphragm. Correlation with the lateral view suggests that this is likely due to
        ↪ interval development of small bilateral pleural effusions. Underlying consolidation is
1197    ↪ not excluded. No pneumothorax. Cardiomediastinal silhouette is otherwise stable.
1198
        IMPRESSION:
1199    Interval development of small bilateral pleural effusions. Underlying consolidation not
1200    ↪ excluded.
1201
        Example 3 (Normal Study):
1202    FINDINGS:
        The lungs are hyperinflated reflective of COPD. Apparent increased opacity projecting over
1203    ↪ the right lung apex correlates with posterior right fifth rib fracture with callus.
1204    ↪ Streaky bibasilar opacities likely reflect atelectasis. No focal consolidation to suggest
        ↪  pneumonia. No pleural effusion or pneumothorax. The heart is normal in size, and the
1205    ↪ mediastinal contours are normal.
1206
        IMPRESSION:
1207    No acute cardiopulmonary process. Focal opacity in the retrocardiac region.
1208
        **REMEMBER**: Do NOT mention attention maps, saliency, heatmaps, or
1209    explainability data. Use model predictions and provided keywords only.
1210
        ### GOAL:
1211    Maximize lexical and semantic similarity to the expert reference report.
1212    Prioritize clinical specificity and exact terminology alignment.
1213
        CHEST X-RAY REPORT:
1214
1215    /no_think
1216    ## Dynamic Components
1217
        ### Condition Predictions Format
1218
        Condition: [CONDITION_NAME]
1219    - Confidence: [XX.X%]
1220    - Clinical Significance: [HIGH/MODERATE/LOW]
        - Keywords: [relevant medical terms]
1221
        ### Clinical Keywords Format
1222
        High Confidence (>80%):
1223    - [keyword1], [keyword2], [keyword3]
1224
        Moderate Confidence (60-80%):
1225    - [keyword4], [keyword5], [keyword6]
1226
        Lower Confidence (40-60%):
1227    - [keyword7], [keyword8], [keyword9]
1228
        ### Anatomical Regions Format
1229
        Primary Focus Areas:
1230    - [anatomical_region_1]: [associated_condition]
1231    - [anatomical_region_2]: [associated_condition]
1232
        Secondary Areas:
1233    - [anatomical_region_3]: [associated_condition]
1234
        ## Report Templates
1235
        ### Standard Template
1236    - **Style:** "professional chest X-ray report"
1237    - **Sections:** ["FINDINGS", "IMPRESSION"]
        - **Length:** Moderate (2-4 sentences per section)
1238
        ### Detailed Template
1239    - **Style:** "comprehensive radiological analysis"
1240    - **Sections:** ["FINDINGS", "IMPRESION", "RECOMMENDATIONS"]
1241    - **Length:** Extensive (4-6 sentences per section)
```

```
### Concise Template
- **Style:** "brief clinical summary"
- **Sections:** ["FINDINGS", "IMPRESSION"]
- **Length:** Brief (1-2 sentences per section)

## Key Safety Features

### Attention Data Prohibition
- **CRITICAL:** No mention of attention maps, saliency, heatmaps, or AI explainability
- Only use model predictions and clinical keywords
- Ensure reports are clinically safe and interpretable

### Confidence-Based Reporting
- High confidence (>70%): Direct reporting
- Moderate confidence (50-70%): Appropriate uncertainty language
- Low confidence (<50%): Do not report finding

### Medical Device Detection
- Always report visible medical devices
- Describe positioning and termination when visible
- Note if device tips are not visible or fully imaged

## Implementation Notes

### Dynamic Variables
- `{reporting_style}`: Determined by case complexity
- `{template_config}`: Based on selected template
- `{condition_predictions}`: Live model outputs
- `{prediction_keywords}`: Extracted clinical keywords
- `{relevant_anatomical_regions}`: Mapped anatomical areas

### LLM Integration
- Compatible with LM Studio API
- Uses `/no_think` directive for direct response - specifically for qwen3
- Optimized for medical report generation models
- Supports both text-only and vision-language models
```

### B.5.2 GROUND TRUTH REPORT (REFERENCE)

For DICOM ID **6c2b39fa-2c251fcf-addd31da-83faee60-044fa8f9**, the expert-provided reference report is:

**Given Report:**

```
                           FINAL REPORT
 INDICATION:  ___F with cirrhosis, recent PNA and alc hep presenting with fever
 101.1 1 week ago.  // evidence of PNA?

 TECHNIQUE:  PA and lateral views the chest.

 COMPARISON:  ___.

 FINDINGS:

 Enteric tube passes below the inferior field of view.  The lungs are clear
 without consolidation, effusion or edema.  The cardiomediastinal silhouette is
 within normal limits.  No acute osseous abnormalities.

 IMPRESSION:

 Enteric tube in place.  No focal consolidation.
```

**Cleaned Report:**
**Findings:** Enteric tube passes below the inferior field of view. The lungs are clear without consolidation, effusion or edema. The cardiomediastinal silhouette is within normal limits. No acute osseous abnormalities.
**Impression:** Enteric tube in place. No focal consolidation.

### B.5.3 MODEL-GENERATED REPORTS

**Gemini 2.5 Pro**
**Findings:** A gastric tube is seen extending into the upper abdomen with its tip likely within the stomach. The lungs are well-aerated. There is no focal consolidation, pleural effusion, or pneumothorax. The cardiac silhouette and mediastinal contours are within normal limits. The visualized osseous structures are unremarkable.
**Impression:** No acute cardiopulmonary process. A gastric tube is in place.

**Qwen3**
**Findings:** The chest radiograph demonstrates aerated lung fields without evidence of focal consolidation or pleural effusion in any zone. The cardiac silhouette is within normal limits. A gastric tube is visualized extending into the upper abdomen; however, the tip is not fully imaged. An additional lead is noted in the upper mediastinum. No acute osseous abnormalities are identified.
**Impression:** No acute cardiopulmonary process. A gastric tube and an additional lead are visualized.

**Llama 4**
**Findings:** The cardiac silhouette appears normal in size. Lung fields are aerated without focal consolidation or pleural effusion. A gastric tube is visible, extending into the upper abdomen, but its tip is not imaged. No acute infiltrates or opacities are seen.
**Impression:** No acute cardiopulmonary process identified. Clinical correlation is recommended regarding support devices.

**MedGemma**
**Findings:** The lungs are well-expanded bilaterally. No evidence of focal consolidation, pleural effusion, or pneumothorax. The cardiac silhouette and mediastinal contours are within normal limits. A gastric tube is visualized extending into the upper abdomen; the tip is not fully imaged. A laparoscopic gastric band is also noted.
**Impression:** No acute cardiopulmonary process. Visible support devices include a gastric tube and a laparoscopic gastric band.

**BioMistral**
**Findings:** A gastric tube is present in the upper abdomen, with the tip not imaged. The lungs are well-expanded bilaterally with no pleural effusion. Cardiomegaly is present. An acute infiltrate is seen in the left upper lung zone and a chronic infiltrate in the right lower lung zone.
**Impression:** Gastric tube present. Cardiomegaly and pulmonary infiltrates. No pleural effusion or pneumothorax.

### B.5.4 COMPARATIVE SUMMARY

Table 16 summarizes the alignment of model outputs with the ground truth report.

| Model | Tube Detection | Lung Findings | Cardiac Findings | Extra/Hallucinated |
|---|---|---|---|---|
| Ground Truth | Yes | Clear | Normal silhouette | None |
| Gemini 2.5 Pro | Yes | Clear | Normal | None |
| Qwen3 | Yes | Clear | Normal | Lead (hallucinated) |
| Llama 4 | Yes | Clear | Normal | Suggests correlation |
| MedGemma | Yes | Clear | Normal | Laparoscopic band |
| BioMistral | Yes | Infiltrates (false) | Cardiomegaly (false) | Multiple findings |

Table 16: Comparison of model-generated reports against ground truth reference.

