# OpenReview forum: "Eyes on the Image: Gaze Supervised Multimodal Learning for Chest X-ray Diagnosis and Report Generation"
_ICLR.cc/2026/Conference — ICLR 2026 Conference Withdrawn Submission_

### Official Review · Reviewer_GhYK · 2025-10-25

**Soundness:** 2
**Presentation:** 2
**Contribution:** 2
**Rating:** 2
**Confidence:** 4

**Summary:**

The paper's direction is significant: coupling human attention with regionalized text generation to pursue interpretable diagnostics. However, further work is needed in areas such as evidence for alignment strength, reliability of regional supervision, factual reporting/reproducibility, and statistical completeness. The paper's persuasiveness would be significantly enhanced if it were supplemented with multi-seed statistics, external validation, open-source backend reproducibility, RadGraph constraints, and YOLO box quality assessment.

**Strengths:**

Highly practical: Gaze alignment is introduced during training, but only images are used during inference. The paper uses a "learnable gaze token + composite alignment loss (MSE/KL/correlation/centroid) + curriculum-based weighting" to constrain attention to human gaze during training. The final test does not rely on any gaze or text input, while still improving discrimination metrics (AUC/F1) and obtaining more interpretable attention maps. This "gaze-for-training, image-only-inference" design is more user-friendly for clinical deployment.

**Weaknesses:**

1. Insufficient distinction between novelty and prior work.
The paper introduces gaze supervision into multimodal chest radiograph classification and report generation, using a core approach of gaze tokens + composite alignment loss + regionalized report generation. However, compared to existing work using gaze for medical representation alignment/attention guidance (e.g., using gaze as weak supervision, as channel/mask input, or as a constraint in multimodal contrastive learning), the methodological incrementality and necessity are insufficiently justified. The related work section primarily provides a side-by-side review, lacking a point-by-point difference table. It is recommended to provide a structural difference diagram/table and a comparison under the same training budget to clarify the need for a learnable gaze token and this specific combination of losses.

2. The interpretation of the "gaze coherence" metric is overly optimistic.
The authors emphasize that the Pearson correlation r≈0.306 achieves "moderate correlation," and the JSD≈0.437 is close to the upper bound for readers. However, the NSS is only 0.138, far below the commonly used human alignment threshold of 1.0. The model's attention entropy also falls short of the upper bound for humans. These observations suggest that "quantitative alignment is weak—visual similarity is insufficient to support strong alignment conclusions." Recommendations include reporting cross-subject/cross-task stratified correlation distributions, introducing Spearman rank correlations/piecewise linear correlations, and statistical significance/confidence intervals to avoid selectively optimistic interpretations of single correlations.

3. Dataset curation and supervision sources may introduce bias. To complete the missing region boxes in REFLACX, the authors trained YOLO using EyeGaze annotation and inferred a maximum confidence box for each region on REFLACX. This was then used for region mapping and report generation. This "self-trained completion" method is prone to mistaking noisy boxes for ground truth, impacting the reliability of region-level supervision and reported geographic attribution. Furthermore, only 2,877 of 3,689 cases (67.1%) were ultimately retained, and this was based on single-center data, raising concerns about generalization. Recommendations include quality assessment of YOLO boxes (intersection of union (IoU)/localization error, sensitivity analysis of subsequent metrics), external data validation, and sensitivity analysis of different thresholds/numbers of candidates (K).

4. Report generation is weak and relies on a proprietary LLM, posing high risks for reproducibility and fairness. Keyword extraction and generation rely on Gemini 2.5 Pro (two-stage filtering and text generation), and the mapping of lesion probabilities to 17 regions is fed to the LLM using a heuristic threshold. However, most text metrics (such as BLEU/ROUGE/METEOR) are low, with RadGraph-F1 at the entity-relationship level being approximately 0.13. The authors omit external horizontal comparisons with existing data/protocols. Uncontrollable updates to the proprietary API also weaken reproducibility and fair comparisons. Recommendations: Provide fully open-source backend reproducible experiments, a list of fixed versions/prompt templates, and comparisons with existing public benchmarks.

5. Potential Impact of Class Selection and Imbalance Treatment
The authors retain eight overrepresented classes and discard minority classes with a combined representation of <1.5%. While this simplification is practical, it may inflate the overall AUC/F1 ratio and obscure the value of gaze for rare disease types. Recommendation: Provide results/recall-cost curves for the minority class scenario and conduct threshold calibration experiments.

6. Figure 2/The "bounded gating layer" in the method description lacks a specific functional form and hyperparameters; the details of the attention rollout resampling/normalization are somewhat vague.

**Questions:**

1. What are the specific forms/hyperparameters of course scheduling and "bounded gating"? Has it been adapted for different departments/readership groups?

2. What is the motivation and anti-leakage measure for "Fine Tune incorporates validation into training"? Is it only used for finalization and does it not change the hyperparameters?

3. What are the quality assessments and failure rates of YOLO boxes in REFLACX? How do the metrics change if the automatic boxes are removed?

4. Are the prompt words, blacklist/whitelist, and regional alias dictionary on the generation side open source? Are the parameters and temperatures of different LLMs consistent?

5. Why not include gaze-text comparisons (the author only uses image-text/image-gaze)? Has this been tried, and what are the results?

---

### Official Review · Reviewer_Q4ZH · 2025-10-28

**Soundness:** 2
**Presentation:** 2
**Contribution:** 3
**Rating:** 4
**Confidence:** 3

**Summary:**

This paper presents an approach to radiology report generation that is trained using multiple modalities of information: the image, gaze data, text report, and bounding boxes. The approach proceeds in two steps, first generating classification labels using a ViT that has features and attention generated from the multimodal input. In the second step, a LLM is used to construct a text report based on the outputs from step 1 as well as a disease/location-specific vocabulary constructed offline.

**Strengths:**

Important problem
- Producing radiology reports that ground findings in correct spatial regions in the image is a high impact problem.

Multimodal approach
- This paper proposes a method to integrate many streams of data with complementary information (the image, the text report, gaze data, and bounding box data) to improve generation performance, which is an interesting approach.
- The particular approach of using gaze data does seem to improve performance compared to not using gaze data and is an interesting way of using gaze to guide image supervision.

**Weaknesses:**

Doubts about the report generation step
- This approach to report generation seems to be more pattern memorizing and pattern matching than true spatial grounding. The authors rely on condition-specific vocabularies (built up offline via processing the MIMIC database with an LLM) to result in phrases that “sound like” radiologists, as well as “ground” those findings using 17 general thoracic regions identified via normalized bounding boxes. While I believe this approach would appear to work well on average (by repeating condition- and region-specific phrases found in the reference MIMIC database), I am not convinced it will generalize or work on edge cases. For example, what if conditions veer from the “typical” vocabulary constructed offline? Is this procedure expected to be repeated at every healthcare institution to match the institute-specific vocabulary? What if conditions appear in abnormal locations or anatomy is abnormal?

Difficult to follow and missing technical details
- I find the organization of the paper confusing, making the method difficult to follow. The main issue I have with the organization is the lack of “big picture” introductions to key parts of the method. The authors present parts of their method using distinct paragraphs with little-to-no explanation for how these distinct parts fit together or what the motivation for them is.  For example:
     - Prior to Section 3.5, there is no motivation given for the offline vocabulary construction, so when this vocabulary construction procedure is described the reader does not yet know how/why this vocabulary will be used. This sort of bottom-up description makes it difficult to follow the method.
     - There are many components to the supervision: contrastive learning, a supervised loss, a gaze-mediated loss. However, these multiple supervision streams are not introduced and contextualized in the Method “Overview and Motivation” section. As a result, when they appear as standalone paragraphs later in the Methods, it is difficult to follow all the different sources of supervision and why they are being used.
     - The authors refer to details about the dataset (e.g., bounding boxes) many times in the Methods section without first describing what those bounding boxes indicate (nor any other details of the dataset), so I don’t know what the bounding boxes are localizing.
     - The section describing the “Enhanced gaze” configuration in 3.2.1 is difficult to parse, particularly because many components rely on equations presented further on in the Methods. For example, the authors refer to using a “composite loss…[to] align A_model with A_gaze”, but it’s not clear what this “alignment” involves until the equation is presented in Section 3.4.
- In addition to difficult organization, there are also many missing technical details, making the method likely very difficult to replicate. If code is not released, I would be unable to replicate this method. Some examples of missing detail are listed below.
     - Curriculum scheduler used for incorporating gaze
     - In line 191, starting “each term…”, how is modulation performed/computed?
     - Precise configurations for each modality encoder
     - In line 172, how is the variable length gaze data first embedded prior to the biGRU?
     - Prompts for vocabulary generation and filtering?
     - In line 209, “otherwise the block is skipped.” When is the block skipped?
     - Are the original images 224 x 224 or are these downsized? If downsized, why? Downsizing is known to reduce accuracy with small abnormalities in medical image interpretation.
     - How are the thoracic regions defined and localized per image?
     - Line 260, “Gemini-cleaned keywords are matched (case-insensitive, fuzzy similarity) to the alias lists.” How is this matching performed, what is the criteria for a successful match?
     - The train/val/test split has percentages that sum to 116.2%, which is not typical for distinct dataset splits.

No baseline methods
- The authors present ablations (comparing their method with and without certain components), but do not compare their method against any of the many other report generation pipelines out there. Without baselines, I cannot assess the strength of the system as a whole nor its ability to meaningfully boost performance/spatial grounding compared to existing solutions.

Typos/grammar/writing
- I think the authors have used \cite{}, which includes citations in-text, where they should have used \citep{}, which includes citations in parentheses. The authors should switch the citations to the correct format as appropriate to improve readability.
- Paragraph 2 assumes a successful report generation system must use gaze data in training, but this is not true.
- Line 236, “Thus 60% of the optimization signal targets label accuracy…” This percentage is not a standard interpretation of loss weights. The scale of each of these losses may be different from one another, so a loss even with a small weight can account for a larger “percentage” of the total loss.

**Questions:**

Addressed in weaknesses.

---

### Official Review · Reviewer_3wAb · 2025-11-01

**Soundness:** 2
**Presentation:** 2
**Contribution:** 2
**Rating:** 4
**Confidence:** 4

**Summary:**

This paper presents a two-stage multimodal framework for chest X-ray interpretation that integrates radiologist eye-tracking data with visual and textual information from the MIMIC-Eye dataset. Stage 1 introduces a gaze-token classifier that fuses image patches, bounding-box masks, transcription embeddings, and radiologist fixations using a trust-calibrated composite loss function. Stage 2 converts classifier predictions into region-grounded diagnostic reports by extracting confidence-weighted keywords, mapping them to 17 thoracic regions, and using an LLM to generate structured reports. The authors report improvements in both classification performance and attention alignment when incorporating gaze supervision.

**Strengths:**

1. The use of radiologist gaze as explicit spatial supervision is innovative and well-motivated. The trust-calibrated composite loss (MSE, KL, Pearson, CoM) is a thoughtful approach to aligning model attention with human expertise.
2. The detailed alignment procedure for MIMIC-Eye demonstrates careful handling of heterogeneous annotations.
3. It provides clear evidence of incremental benefits from each modality, and the extended ablations examine backbone choices systematically.
4. Comparing six different LLMs provides valuable insights into generation quality trade-offs.

**Weaknesses:**

1. The entire evaluation is on a single dataset (MIMIC-Eye, n=2,877 aligned samples). It lacks comparison with state-of-the-art on standard benchmarks.
2. When comparing report generation performance, the author does not compare their methods with current report generation models.
3. This method introduces many modules for intermediate fixation prediction but they do not compare the computational efficiency of their method and current report generation methods.
4. I think this method is somehow cascaded several modules, while these modules cannot show their novelty.

**Questions:**

see Weaknesses

---

### Official Review · Reviewer_YLVC · 2025-11-01

**Soundness:** 2
**Presentation:** 1
**Contribution:** 2
**Rating:** 4
**Confidence:** 4

**Summary:**

This work proposes a new gaze-guided, region-grounded framework for interpreting CXRs that combines visual, textual, and attentional modalities is presented in this paper. The proposed work aims to improve interpretability and report accuracy by using human fixation data for spatial supervision. The method advances medical AI reporting and has potential for clinical applicability.

**Strengths:**

The main strengths of the paper are that it presents a novel multimodal framework that combines eye-tracking data, radiology text, anatomical masks, and chest X-ray images. The two-stage design, which combines region-grounded text generation and gaze-guided classification, is both conceptually and clinically sound. The methodological rigor of the proposed method is further reinforced by thorough ablations and meticulously documented dataset curation.

**Weaknesses:**

Major:
1. The main flaw is that I fail to see how this approach advances clinical practice. Annotations are even more challenging than finding medical data to train AI models.  It might not be feasible to design an eye gaze experiment to regulate the annotations in the suggested method.
2. One of the major concerns is the experimental validation of the proposed method. A few points that are lacking: a) an experiment that uses shuffled/mismatched transcripts or eliminates text from Stage 1 (and from any pretraining that touches the same split); b) a label-only text control (text→labels logistic baseline) to measure the amount of signal derived from text alone during training; and c) leakage checks to ensure that the text used in contrastive learning does not contain templated CheXpert phrases that mirror targets.
3. The Introduction section of the paper, which primarily discusses the problem statement, clinical motivation, and hypothesis of the proposed work, is poorly written. As a result, the paper becomes challenging to understand.
4. The authors should show some examples of generated reports and ground truth reports as qualitative results.
5. “Region grounding” is asserted but not convincingly validated. Region signals come from (i) EyeGaze boxes, (ii) YOLO-predicted boxes for REFLACX, (iii) keyword→region alias mapping at inference. Specially in case of REFLACX, if keyword→region mapping and report prompts depend on YOLO-generated regions, the text may be steered by potentially incorrect regions, confounding the interpretation of generator scores and any “grounding” visualization.
6. Also in experimental results, surface metrics like BLEU, ROUGE, METEOR, etc. are low across LLMs. The RadGraph-F1 score is ~0.13 (Table 3), which is modest and directly contradicts the claim of “entity-level” grounding.
7. The experiments lack statistical validation. Without that, it is insufficient to validate the key results. It would also help if authors add confidence intervals to the results.

Minor:
1. Hyperparameter selection is not discussed. For example, in Eq. 6, how $\lambda_{NCE}$ and $\lambda_{gaze}$ are selected.
2. The attention map is referred to as the attention rollout. Rollout can blur at deeper layers, so it is important to compare alternative techniques (LRP, Grad-CAM, etc).
3. Several mathematical notations are not consistent. Such as loss functions are represented as $\mathcal{L}$ and $L$ multiple times.

**Questions:**

See weakness section.

---

### Note · Authors · 2025-11-28

I have read and agree with the venue's withdrawal policy on behalf of myself and my co-authors.